# Influence of PEG Chain Length of Functionalized Magnetic Nanoparticles on the Cytocompatibility and Immune Competence of Primary Murine Macrophages and Dendritic Cells

**DOI:** 10.3390/ijms24032565

**Published:** 2023-01-29

**Authors:** Ronja Storjohann, Birthe Gericke, Janin Reifenrath, Timo Herrmann, Peter Behrens, Hilke Oltmanns, Jessica Meißner

**Affiliations:** 1Department of Pharmacology, Toxicology and Pharmacy, University of Veterinary Medicine Hannover, Foundation, Buenteweg 17, 30559 Hannover, Germany; 2Center for Systems Neuroscience, 30559 Hannover, Germany; 3Hannover Medical School, Clinic for Orthopedic Surgery, Carl-Neuberg-Straße 1, 30625 Hannover, Germany; 4Lower Saxony Centre for Biomedical Engineering, Implant Research and Development (NIFE), Hannover Medical School, Stadtfelddamm 34, 30625 Hannover, Germany; 5Institute of Inorganic Chemistry, Leibniz University Hannover, Callinstraße 9, 30167 Hannover, Germany

**Keywords:** superparamagnetic iron oxide nanoparticles, nanoporous silica nanoparticles, phagocytosis, targeted drug delivery, biocompatibility, immunotoxicology, Fe_3_O_4_

## Abstract

A major drawback of nanoparticles (NPs) for biomedical applications is their preferential phagocytosis in immune cells, which can be avoided by surface modifications like PEGylation. Nevertheless, examinations of different polyethylene glycol (PEG) chain lengths on the competence of immune cells as well as possible immunotoxic effects are still sparse. Therefore, primary murine macrophages and dendritic cells were generated and incubated with magnetic nanoporous silica nanoparticles (MNPSNPs) modified with different mPEG chains (2 kDa, 5 kDa, and 10 kDa). Cytotoxicity, cytokine release, and the formation of reactive oxygen species (ROS) were determined. Immune competence of both cell types was examined and uptake of MNPSNPs into macrophages was visualized. Concentrations up to 150 µg/mL MNPSNPs showed no effects on the metabolic activity or immune competence of both cell types. However, ROS significantly increased in macrophages incubated with larger PEG chains, while the concentration of cytokines (TNF-α and IL-6) did not indicate a proinflammatory process. Investigations on the uptake of MNPSNPs revealed no differences in the onset of internalization and the intensity of intracellular fluorescence. The study gives no indication for an immunotoxic effect of PEGylated MNPSNPs. Nevertheless, there is still a need for optimization regarding their internalization to ensure an efficient drug delivery.

## 1. Introduction

Nanoparticles (NPs), especially magnetic ones, are popular in biomedicine due to their high biocompatibility [1,2,3,4,5]. Different NPs are already used in research and in the clinic for diagnostic purposes [6,7,8], in hyperthermia [9,10], as well as drug delivery in tumor and infection treatment [5,11,12,13,14,15]. The ability to load them with several substances, such as antibiotics, and thus the possibility of targeted drug administration can be a promising alternative to systemic drug treatment. However, due to rapid phagocytosis in the bloodstream and tissues by the mononuclear phagocyte system [5,16,17,18], the number of NPs reaching their target tissue is diminished. Therefore, several surface modifications have been investigated [18,19,20]. Besides modifications with polyvinyl alcohol [21] and dextran [22], surface modification with polyethylene glycol (PEG) has been investigated to avoid phagocytosis into cells [19,20,23]. Thus, a prolonged circulation time in the bloodstream was observed to enhance the number of NPs reaching tumors [24]. The influence of PEGylation on the circulation time seems to improve with increasing PEG mass [20]. Likewise, other parameters, such as the particle diameter and the coating density, affect the stealth effect [24].

PEG is generally considered non-immunogenic [25]; nevertheless, anti-PEG antibodies or at least allergenic reactions have been reported [26,27,28]. Studies dealing with the effects of different PEG chain masses on the functions of the immune system are sparse, especially in light of iron oxide NPs designed for drug delivery.

The present study investigated the impact of different mPEG chains (2 kDa, 5 kDa, and 10 kDa) used for surface modification of magnetic nanoporous silica nanoparticles (MNPSNPs) on different immune cells in vitro. The underlying first hypothesis was a that there is a reduction in MNPSNP uptake by macrophages with an increasing mPEG mass. The second hypothesis was that there is no change in the biocompatibility of the particles with changes in mPEG mass. A conceivable reaction of the immunocompetent cells towards the MNPSNPs was of great interest to evaluate their stealth effect. Therefore, we evaluated different parameters like metabolic activity, cytokine secretion, production of reactive oxygen species (ROS), and specific immune related processes of the cells. In addition, the effect of PEGylation on the internalization by macrophages was investigated as a central issue.

## 2. Results

### 2.1. Characterization of Unfunctionalized and Functionalized MNPSNPs

The synthesized spherical MNPSNPs showed a uniform size distribution with a diameter of 123 ± 16 nm, as determined by transmission electron microscopy (TEM). Dark spots could be seen within the MNPSNPs, which represent the magnetite cores. These cores showed a diameter of 10 ± 2 nm. The shape and pore system of the particles after functionalization were preserved, as shown in Figure 1.

The surface properties of the PEGylated MNPSNPs were investigated using nitrogen physisorption. In Figure 2, the physisorption isotherms were shown for the different mPEG functionalization, as well as for the unfunctionalized particles. The attachment of the mPEG cloud be confirmed while a high porosity and a large pore volume was maintained, which is needed for the desired drug delivery application.

In general, a decrease in the Brunauer–Emmett–Teller (BET) surface area with an increasing amount of mPEG was observed. The BET surface area decreased from 910 m^2^·g^−1^ for the unfunctionalized MNPSNPs to 730 m^2^·g^−1^, 610 m^2^·g^−1^, and 640 m^2^·g^−1^ for the 2 kDa mPEG, 5 kDa mPEG, and 10 kDa mPEG, respectively. Noticeably, when increasing the chain length from 5 kDa to 10 kDa mPEG, no further decrease in the BET surface was observed when compared to changing the length from 2 kDa to 5 kDa mPEG. A similar trend was observed in the pore diameter and pore volume, as shown in Table 1.

The amount of organic material bound to the surface of the MNPSNPs was further investigated by thermogravimetric analysis (TGA), as shown in Figure 3. During the TGA, the organic material attached to the particles was decomposed and water was released by condensation of the silanol groups, leaving the inorganic material. As a result, conclusions about the amount of organic material bound to the particle surface could be made; a lower residual mass denotes a higher organic content. It could be demonstrated that the residual mass of the MNPSNPs was decreased as the mPEG chain length increased. Unfunctionalized MNPSNPs showed a mass loss of 6% caused by the loss of adsorbed water and the dehydroxylation of the silanol groups. The mass loss of the functionalized MNPSNPs increased with increasing mPEG chain length from 14% for the 2 kDa mPEG functionalization to 18% and 20% for the 5 kDa mPEG and 10 kDa mPEG functionalization, respectively. That being the case, the particle mass also increased with the mPEG chain length when the same amount of the respective mPEG was bound to the surface.

### 2.2. Cytotoxicity of Functionalized MNPSNPs

The cytotoxicity of PEGylated MNPSNPs on the immune cells were analyzed via the MTS assay in primary murine macrophages and dendritic cells (DCs) (Figure 4). After an incubation period of 24 and 48 h, both showed no significant reduction in metabolic activity. In DCs, an increased absorbance could be seen, which positively correlates with the concentration of MNPSNPs.

### 2.3. Effect of Functionalized MNPSNPs on Cytokine Release

Further evaluations of the proinflammatory effect of MNPSNPs on the immune cells were conducted by analyzing the culture supernatants for cytokines using enzyme-linked immunosorbent assays (ELISAs) (Figure 5). While macrophages showed a significantly increased secretion of tumor necrosis factor-α (TNF-α) after treatment with 10 kDa mPEG-MNPSNPs (50 µg/mL) for 24 h, the release in DCs increased after 24 h of incubation with 2 kDa mPEG-MNPSNPs (50 µg/mL) and 5 kDa mPEG-MNPSNPs (15 and 150 µg/mL). In both cases, no significant effect of MNPSNP-treatment could be observed after 48 h of incubation. In addition, the positive control showed a more pronounced effect on the cytokine release.

The murine macrophages secreted TNF-α at a median of 5.7 pg/mL, which elevated to 9.5 pg/mL after treatment with 10 kDa mPEG-MNPSNPs (50 µg/mL). In contrast, lipopolysaccharide (LPS)-treatment increased the cytokine concentration by a factor of 523 to 2981 pg/mL in the median. Similarly, DCs secretion of TNF-α showed a median of 67 pg/mL for control, which elevated to 97 pg/mL (2 kDa mPEG-MNPSNPs) or 86.6 pg/mL and 100 pg/mL (5 kDa mPEG-MNPSNPs). Again, the positive control with LPS-stimulation increased the greatest to 12,347 pg/mL by a factor of 184.

The investigations on the cytokine interleukin 6 (IL-6) showed a comparable effect of MNPSNPs in the immune cells studied (Figure 6). While macrophages showed a significant increase in secretion after 24 h of treatment with 150 µg/mL 2 kDa mPEG-MNPSNPs (median: 1.4 pg/mL, factor 2), no effects were seen in DCs. As for TNF-α, the macrophage response to the MNPSNPs remained markedly below the positive control (15,885 pg/mL, factor 22,693). No effect was seen after 48 h.

### 2.4. Effect of Functionalized MNPSNPs on the Formation of Reactive Oxygen Species

The formation of intracellular ROS was measured using a 2′,7′ dichlorofluorescein diacetate (DCFDA) assay (Figure 7A,B). The incubation of macrophages with the different MNPSNPs for 24 h led to a significant increase of ROS compared to the control. Moreover, intracellular ROS formation in macrophages tended to show a positive correlation with mPEG mass. Whereas no effect of the MNPSNPs on the DCs was observed.

### 2.5. Impact of Functionalized MNPSNPs on the Cell Migration of DCs

To investigate the emigration behavior of the DCs, the cells were incubated in a transwell system (Boyden chamber) with different MNPSNPs for 90 min. There was no significant increase of the cell number in the lower compartment, neither for MNPSNPs nor for the chemokine macrophage inflammatory protein-3β (MIP-3β) (Figure 7C). However, a slight tendency to increased migration could be shown for MIP-3β.

### 2.6. Impact of Functionalized MNPSNPs on the Immune Competence of Macrophages

The ability of macrophages to take up and kill a pathogen was tested by infection with *Escherichia coli* (*E. coli*) following MNPSNP treatment. Thereby, the treated macrophages did not show significantly enhanced loss of function compared to control (Figure 7D).

### 2.7. Intracellular Uptake of Functionalized MNPSNPs

To investigate the effect of PEGylation on the particle internalization by immune cells, the different MNPSNPs were incubated with primary murine macrophages for various intervals. The fluorescence microscopic images showed an increase in corrected intracellular fluorescence over the time independent of mPEG mass. No differences were observed in the internalization time and intensity of intracellular fluorescence between the different MNPSNPs (Figure 7E). After 5 min of incubation, the MNPSNPs were primarily detected on the cell surface with only a few isolated particles enclosed by the cell membrane (Figure A1). Continuing, after 10 min, the first particles also appeared intracellularly, while 20 min of incubation resulted in increased uptake of MNPSNPs by macrophages. The interval of 90 min, which was the longest examined, showed a clear accumulation of MNPSNPs intracellularly and a persistent internalization could be seen (Figure A1). Furthermore, images of the XY- and XZ-plane, showed no intranuclear MNPSNPs (Figure A2).

## 3. Discussion

Most studies on NPs are focused on the intended uptake by the targeted cells. However, for some treatments, free circulation and extravasation at the site of infection is preferable. It has been shown that a sufficiently long circulation of MNPSNPs is mainly affected by the cells of the mononuclear phagocyte system [16]. Therefore, the present study was designed to investigate the influence of mPEG functionalization of MNPSNPs on macrophages and DCs. Immunotoxic effects were examined by metabolic activity, cytokine secretion, ROS production, and immune competence, while the internalization of MNPSNPs was studied comparatively for the different mPEG chains.

As expected, the metabolic activity of the macrophages and DCs showed no significant reduction in cell viability after 24 and 48 h of treatment with the tested MNPSNPs up to a concentration of 150 µg/mL (Figure 4). Taking into account the ISO 10993-5:2009-06 for medical devices [29], the MNPSNPs were found to be non-cytotoxic. Further, only low levels of the cytokines TNF-α and IL-6 were measured in the culture supernatants (Figure 5 and Figure 6). On the one hand, these were significant compared to the untreated control, but on the other hand, TNF-α and IL-6 cytokine levels were markedly lower than the secretion caused by stimulation of the cells with LPS (positive control). The slight increase in proinflammatory cytokines was not seen to be related to a proinflammatory response when particles were used in vivo. The biocompatibility examinations could confirm that none of the PEG chains were noticeably worse than the others. These results are in accordance with previous studies on MNPSNPs without further surface functionalization [16,30]. While MNPSNPs showed good biocompatibility in murine fibroblasts (NIH-3T3) and human hepatoma cells (HepG2) [30], no significant pathomorphological changes were seen in mice 42 days post-injection [16]. The observed increase of metabolic activity in the DCs could not be associated to an immune response in further experiments. In addition, there was no evidence for an uncoupling of the respiratory chain, as described for astrocytes, after incubation with iron oxide-containing NPs [31], which could cause an inaccuracy in the measurements of metabolic activity. Such an imbalance of electron transport and the proton gradient might also lead to ROS formation due to a decrease in ATP production [32], which was not observed in this study. Likewise, described interference with mitochondrial function and a resulting inaccuracy in the measurements [33,34], as well as an artificial detection of MNPSNPs in the absorption readings, are excluded due to an absent effect in the experiments with macrophages and the background controls carried along (Table A1, Table A2, Table A3, Table A4, Table A5 and Table A6).

A study that examined silica-coated Fe_3_O_4_-NPs (Ø 30, 50 and 120 nm, −22 mV) in macrophages showed a trend towards reduced viability with increasing NP concentration, whereas no enhancement of proinflammatory cytokine levels could be detected [35]. Human DCs also showed no cytotoxic effect up to a concentration of 100 µg/mL, and measurements of the proinflammatory cytokines detected neither TNF-α nor IL-6 in significantly increased amounts [35]. In contrast, silica NPs without an iron oxide core can decrease the metabolic activity of DCs in a dose- and size-dependent manner and, especially small particles (Ø 20 nm), can lead to increased TNF-α secretion [36]. In addition, dextran-coated NPs showed a proinflammatory effect on human monocytes, increasing both TNF-α and IL-6 [37]. The influence of coating and surface charge were further demonstrated in a comparison of PEGylated and polyethylenimine-coated NPs, where the latter were found to be more toxic due to a positive zeta potential [38]. No differences in zeta potential were seen for the MNPSNPs examined, but a slight decrease occurred due to the binding of the fluorochrome FITC (Table 1).

It has been shown that some cells respond to NPs with increased ROS production. However, this may vary depending on the individual particle material and structure, as well as the cell type investigated [38,39,40,41,42,43]. Therefore, this study examined intracellular ROS development in macrophages and DCs after 24 h of incubation, in addition to the parameters already mentioned. Deviating from most protocols, a longer reaction time with the test substance DCFDA of 4 h was chosen for this purpose to strengthen the differences between the untreated and treated cells. Nevertheless, no significant increase in intracellular ROS could be measured in the DCs compared to control (Figure 7B). In contrast, macrophages showed a significant enhancement in ROS generation, which tended to progress with heavier mPEG chains (Figure 7A). An increased production of ROS is suggested to be associated with the degradation of iron within the cell and a further reaction with surrounding hydrogen peroxide (Fenton-Reaction) [42,43]. In this context, silica-coated iron oxide NPs without further surface functionalization showed a significant effect on the ROS production in adenocarcinomic cells (A549) und cervical cancer cells (HeLa) at lower concentrations than those particles with additional surface passivation by sulfate or amino groups [42]. Furthermore, the administration of an iron chelator improved the viability of the cells [42]. A slower or even negligible degradation of iron oxide due to the thick silica layer and functionalization of the particle surface would be conceivable for the investigated MNPSNPs and a possible explanation for the good tolerance in DCs. Nevertheless, this hypothesis would contradict the observed trend of increasing ROS levels with higher mPEG mass in the macrophages. The impact of a higher ROS production varies and reflects the complexity of the underlying mechanisms. Although different iron oxide NPs increased the intracellular ROS levels in A549 cells, no DNA damage or elevation of total antioxidant capacity could be detected [43]. In addition to the cellular factors, the construction of the particles was also shown to be crucial. As an example, a non-porous silica surface increased ROS production in murine macrophages more than a porous particle surface [44]. The experiments on MNPSNPs in macrophages showed increasing ROS levels, which correlated positively with mPEG mass. However, no evidence for decreased viability was found in the MTS assay. This may indicate that ROS were sufficiently neutralized by the cellular antioxidant system. Further studies on DNA damage or the various antioxidants and enzymes, as well as the iron regulation genes, could give a more detailed picture.

This study aimed to consider the effects on immune competence in addition to the more commonly examined parameters like viability and cytokine secretion. First, the chemotactic potential of MNPSNPs on DCs was tested. A suppressing effect on the migration, as described for single-walled carbon nanotubes in human monocytes [45], could not be observed. While the MNPSNPs did not seem to provide an adequate chemotactic stimulus, the compartments containing MIP-3β tended to contain a higher number of DCs (Figure 7B). The small effect of the positive control can probably be explained by the sparse number of maturated DCs with the appropriate receptor (CCchemokine-receptor 7) [46,47,48].

Further, the described infection trial demonstrated that macrophages maintained the ability to kill internalized *E. coli* even after contact with and presumed uptake of MNPSNPs (Figure 7D). Since no significant effect of previous incubation with iron oxide NPs on pathogen uptake was observed [37], it can be assumed that the ingestion of *E. coli* remains unchanged. In addition to the immune competence of the cell, possible antibacterial properties of iron oxide NPs described in the literature should also be discussed [49]. Besides an effective elimination of *E. coli* by the macrophages, intracellular contact of the bacteria with the particles seems to be possible, resulting in an antibacterial effect. However, this is considered unlikely due to the silica shell and PEGylation, as well as the observed minimal bacterial growth. A further reduction of colony forming units below the untreated control was not observed. Likewise, survival of non-internalized viable *E. coli* seems unlikely.

As mentioned above, the internalization of MNPSNPs by the cells of the mononuclear phagocyte system is the main obstacle to accumulate a sufficiently number of particles [16]. Therefore, the MNPSNPs were functionalized with different mPEG chains and comparatively incubated with macrophages. Since the Prussian blue staining for the detection of internalized iron-containing particles could not be implemented due to a required demasking with sodium hydroxide solution [50], fluorescence microscopy was chosen alternatively. It was decided to scan the images in the XY-plane, in which a distinction between intra- and extracellular MNPSNPs was generally possible. To verify its accuracy, the XZ-plane was occasionally observed, and exemplary Z-stack images were taken (not shown). Defining the intracellular area manually differs from other approaches using image analysis program algorithms [51], but allowed exclusion of non-internalized MNPSNPs on the cell surface. Despite this, the detection of strongly fluorescent particles on the lower side of the cells or invaginated cell surfaces cannot be completely ruled out. While comparable protocols analyzed 30 or 150–500 cells [51,52], this study evaluated fewer cells per setup. Likewise, it should be mentioned that due to the compounding process with simultaneous binding of mPEG chains and fluorescein-isothiocyanat isomer I (FITC), a differing fluorochrome loading of the MNPSNPs could not be excluded. However, the experiments are suitable for an initial comparison regarding the efficiency of the different mPEG chains on the internalization time of the MNPSNPs. The observed internalization was time-dependent and showed a comparably fast uptake into the cells for all MNPSNPs. Over the duration of 90 min, there was a continuous uptake of MNPSNPs. Further, MNPSNPs were not seen in the nucleus at any time. A lengthening of the half-life with increasing PEG mass, as described both in vitro and in vivo for different NPs [19,20,23,24,38], could not be confirmed for the investigated MNPSNPs.

A study on gold NPs with different diameters (Ø 17.72–86.73 nm) investigated PEG chains of the same molar masses (2, 5, and 10 kDa). Here, NPs with a heavier chain showed a prolonged half-life in mice at constant diameter (Ø 86.73 nm: 3.3 h (5 kDa), 6.6 h (10 kDa)), while larger particles were eliminated faster at constant PEGylation (10 kDa: 51.1 h (Ø 17.72), 6.6 h (Ø 86.73)) [20]. In addition, the coating density is another parameter to be considered, since it changes the distance between the PEG chains and the fixed aqueous layer thickness (FALT) properties [24]. NPs showed a thicker FALT with increasing PEG mass and the same diameter or with decreasing diameter and the same PEG mass. Increasing PEG mass was accompanied by a wider distance of the chains to each other, which in turn may have an effect on PEG chain conformation and protein adsorption [24]. Thus, for the poly methoxyPEG cyanoacrylate-co-n-hexadecyl cyanoacrylate NPs with a size of 80 nm and a 5 kDa PEG chain, the longest half-life, highest tumor accumulation, as well as lowest protein adsorption and accumulation in the liver were observed in vitro and in vivo [24]. Since the MNPSNPs were prepared by the same method until PEGylation, the core size and the silica shell should not differ. While the diameter of MNPSNPs with and without mPEG were not found to be different in TEM, previous studies showed a variable increase in hydrodynamic diameter [30]. Indeed, different PEG weights may lead to different hydrodynamic diameters in the following [20], suggesting that a possible masking of effectiveness by resulting different diameters could be conceivable. To support this hypothesis, further characterization of the diameters, the FALT, and protein adsorption would be necessary.

In addition to the described effects on the half-life, PEG promotes the monodispersity of NPs [38,40,53]. It has been shown that silica NPs are incorporated by different mechanisms and in various quantities, depending on their isolated or agglomerated state. In this regard, an agglomerated form promoted their uptake [54]. In the fluorescent microscopic experiments, agglomerates were observed for all MNPSNPs, so it cannot be ruled out that increased uptake resulted. However, the mass concentrations of MNPSNPs were kept constant for all tests and thus allowed comparisons among them. Interactions between the positively charged poly-L-lysine coating of the coverslips and the negatively charged MNPSNPs may have led to increased retention of the particles during the washing steps. However, due to the subsequent fixation of the cells, this should not have caused any inaccuracies in the measurements.

Besides further optimization of the PEGylation by adjustments of the coating density, hydrodynamic diameter, or zeta potential, other methods are described and offer a potential alternative to PEG [55,56]. Most notably, binding of the CD47 protein as a self-marker could extend the half-life of the MNPSNPs to allow accumulation. In this regard, initial promising tests have already been carried out in the course of the project and point out a potential way forward for the MNPSNPs.

## 4. Materials and Methods

### 4.1. Materials

For the syntheses, all chemicals were used without further purification. Iron(II) chloride tetrahydrate (≥99%), iron(III) chloride tetrahydrate (99%), oleic acid (90%), chloroform (≥99%), cetyltrimethylammonium bromide (CTAB, ≥98%), ammonium hydroxide solution (≥25% NH_3_ in water), tetraethyl orthosilicate (TEOS, ≥99%), ethyl acetate (99.8%), (3-aminopropyl)triethoxysilane (APTES, 99%), (3-aminopropyl)trimethoxysilan (APTMS, 97%), ethanol (Absolute, Emplura^®^, Merck KGaA, Darmstadt, Germany), poly(ethylene glycol) monomethylether (5 kDa mPEG, M_w_: 5 kDa; 10 kDa mPEG, M_w_: 10 kDa), sodium hydroxide (NaOH, Emsure^®^, Merck KGaA, Darmstadt, Germany), hydrochloric acid (HCl, 2 M), and Fluorescein-isothiocyanat Isomer I (FITC, ≥99%) were purchased from Sigma–Aldrich Corporation (München, Germany); poly(ethylene glycol) monomethylether (2 kDa mPEG, M_w_: 2 kDa) was purchased from TCI Europe N.V. (Zwijindrecht, Belgium); tosylchloride (98%) and tetrahydrofuran (THF, 99.8%) were purchased from Acros Organics (Geel, Belgium); dichloromethane (DCM, ≥99.9%) was purchased from Honeywell GmbH (Seelze, Germany); and MgSO_4_ (99%) was purchased from ThermoFisher (Kandel) GmbH (Kandel, Germany).

In vitro experiments were performed in ROTI^®^Cell Roswell Park Memorial Institute Medium (RPMI-1640) with or without phenol-red purchased from Carl Roth GmbH & Co. KG (Karlsruhe, Germany). Fetal calf serum (FCS) was purchased from Biochrom GmbH (Berlin, Germany) and Biological Industries Israel Beit-Haemek (Kibbutz Beit-Haemek, Israel). Penicillin (10,000 U/mL)/Streptomycin (10,000 μg/mL) was purchased from Gibco Life Technologies Corporation (Grans Island, USA), while lipopolysaccharide (LPS, *Escherichia coli* O127:B8) and β-mercaptoethanol were purchased from Sigma–Aldrich Chemie GmbH (Steinheim, Germany).

For the Schwinzer erythrocyte lysis buffer, 8.3 g ammoniumchloride (Merck KgAa, Darmstadt, Germany), 0.1 g EDTA (Sigma-Aldrich Co., St. Louis, MO, USA), and 1.0 g KHCO_3_ (Merck KgAa, Darmstadt, Germany) were dissolved in 100 mL *aqua bidestillata*.

The M9 medium was prepared from 5 g/L D(+)-glucose-monohydrate (Merck KgAa, Darmstadt, Germany), 6 g/L dinatriumhydrogenphosphate-anhydrate (AppliChem GmbH, Darmstadt, Germany), 3 g/L kaliumdihydrogenphosphate (Merck KgAa, Darmstadt, Germany), 1 g/L ammoniumchloride (Merck KgAa, Darmstadt, Germany), 0.5 g/L natriumchloride (Merck KgAa, Darmstadt, Germany), 0.12 g/L magnesiumsulfate (Merck KgAa, Darmstadt, Germany), 0.01 g/L calciumchloride-dihydrate (Merck KgAa, Darmstadt, Germany), and 1 mL/L thiamine-hydrochloride (20 mg/mL, AppliChem, Darmstadt, Germany) in *aqua bidestillata*.

### 4.2. Synthesis of Magnetic Nanoporous Silica Nanoparticles (MNPSNPs) and Functionalization with mPEG

#### 4.2.1. Synthesis of MNPSNPs

For the synthesis of MNPSNPs, hydrophobic magnetite NPs were firstly prepared by the same method as previously reported [16,30]. Next, a porous silica shell was condensed around the magnetite NPs, using the synthesis reported in the literature with slight modifications [16,30]. Briefly, the magnetite NPs were transferred into the aqueous phase using the surfactant CTAB. Next, water was added, and the dispersion was stirred for 30 min at 60 °C. Afterwards, ammonium hydroxide, TEOS, and ethyl acetate were added, and the reaction mixture was allowed to react for 3 h at 60 °C. After cooling, the light brown product was magnetically separated, centrifuged at 6000× *g*, and washed three times with 10 mL of ethanol. The surfactant was then removed by calcination at 550 °C for 5 h with a heating rate of 1 °C/minute.

#### 4.2.2. Synthesis of mPEG Silanes with Different Molecular Weights

The synthesis of the mPEG silane was carried out using a two-step synthesis following a published procedure with some minor changes [57]. Depending on the mass of the mPEG derivative, 6 mmol, 3 mmol, or 0.5 mmol were used for the 2 kDa mPEG, 5 kDa mPEG, and 10 kDa mPEG, respectively. In general, the mPEG derivative was dissolved in a mixture of THF and 0.54 M NaOH (2 kDa mPEG: 10 mL THF, 40 mL 0.54 M NaOH (1:4), 5 kDa mPEG: 1:2, 10 kDa mPEG: 1:0.3). The volume of the NaOH solution was chosen in such way that a 3.6-fold excess of NaOH in relation to the amount of mPEG was achieved. The solution was allowed to stir for 30 min at 0 °C and a clear solution was obtained. Next, a 1.2 molar excess of tosylchloride was dissolved in 10 mL THF and added dropwise within 45 min to the mPEG solution. After the complete addition, the solution was further stirred for 3 h at 0 °C. Next, the solution was poured into 50 mL of 1 M HCl and briefly shaken. The product was extracted three times using 50 mL DCM, dried with MgSO_4_ and filtered. The solvent was removed, and the product was dried under a vacuum. The crude product was used for the next step without further purification. Reaction was carried out with an equimolar amount of APTES in 25 mL chloroform for 17 h under reflux. To obtain the final product, the solvent was removed and the product was dried under vacuum. The obtained mPEG silane was stored as a 50 *wt*% solution in ethanol at 4 °C and used for the functionalization of the MNPSNPs without further purification.

#### 4.2.3. Functionalization of MNPSNPs with mPEG and FITC

For the FITC labeling of the MNPSNPs, a FITC silane was synthesized by reacting 18 mg FITC with 5.4 µL APTMS in 2 mL ethanol for 24 h in the dark [30,58]. For the functionalization, 120 mg MNPSNPs were dispersed in 42 mL ethanol. Next, 0.725 mmol of the desired mPEG silane and 502 µL of the FITC silane solution were added and the solution was stirred at 50 °C for 24 h under the exclusion of light. The functionalized MNPSNPs were obtained by centrifugation, washed with copious ethanol, and dried under a vacuum. The obtained particles are denoted as MNPSNP-xkDa mPEG-FITC, where *xkDa* indicates the respective mean molar mass of the mPEG in thousands.

### 4.3. Characterizations

TEM was performed using a FEI Tecnai F20 TMP (*C*_S_ = 2 mm, *C*_C_ = 2 mm) with a 200 kV emission field gun. For preparation, the sample was dispersed via ultrasonication, dropped on a 400-mesh copper grid (plano GmbH, Wetzlar, Germany), and dried. The particle size was determined using NIH ImageJ. Physisorption measurements were performed on a Quantachrome Autosorb-3 instrument with N_2_ as adsorptive. The sample was outgassed for 24 h at 110 °C. Surface areas were calculated using the software ASiQwin (version 2.0) from Quantachrome using BET equation. Pore sizes were calculated applying the density functional theory, where the experimental data were fitted to the Kernel N_2_ at 77 K on silica (cylinder/sphere pore, NLDFT ads. Model) from the Quantachrome software. The total pore volume was determined with the single point method at *p/p*_0_ = 0.95. Thermogravimetric measurements were performed on a Netzsch STA 449 F5 Jupiter. The samples were heated to 1000 °C with a rate of 5 °C/minute in a Al_2_O_3_ crucible. The measurement was performed in a mixed atmosphere of N_2_ and O_2_ (80%:20%). Zeta potentials were measured by dynamic light scattering (DLS) with the Zetasizer Nano ZSP (Malvern Panalytical), wherefore the sample was dispersed in water.

### 4.4. Animals

Husbandry and handling of mice used for the cell generation was done according to German animal welfare regulations. Fifteen male wild-type mice of the BALB/c line were used at the age of 8–12 weeks. All animals were obtained from Charles River (Sulzfeld, Germany). The husbandry took place in an enriched macrolon cage type III in groups of 2–6 animals. Nest building material and a tube were presented as enrichment measures. With a light-dark cycle of 12 h and a room temperature of 22 °C, the animals got food and water ad libitum. The displayed project had the file number TiHo-T-2020-10.

### 4.5. Generation of Primary Immune Cells

Primary macrophages and DCs were generated according to Lutz et al. (1999), as described by Bäumer et al. (2003) [46,47]. Mice were euthanized by cervical dislocation and then wetted with 70% ethanol. Under sterile conditions, the Os femoris was dissected (from both sides) with forceps and a scalpel blade and transferred to a petri dish with warmed, sterile phosphate buffered saline (PBS) (36 °C). Muscles and tendons were removed from the bones before they were disinfected in 70% ethanol for a few minutes. After transfer to a petri dish filled with PBS, attached tissue was removed from the bones as well as possible. The epiphyses were severed with the scalpel blade and the bone marrow was flushed out into a 50 mL centrifuge tube using a cannula (25 G), a disposable syringe, and warmed PBS. The bone marrow was centrifuged for 5 min at 290× *g* and room temperature (RT) and the cells were then resuspended in 1 mL of prewarmed DC-medium consisting of RPMI-1640 with 10% fetal calf serum (FCS), 1% penicillin/streptomycin and 50 µM β-mercaptoethanol. After adding 4 mL of medium, 100 μL of the cell suspension were incubated with 100 μL of Schwinzer erythrocyte lysis buffer for 5 min at RT.

Three million progenitor cells were placed in 10 mL DC-medium on a tissue culture dish (100 mm Cell+, Sarstedt AG & Co. KG, Nümbrecht, Germany) and were, after adding 20 ng/mL recombinant mouse granulocyte macrophage colony-stimulating factor (rmGM-CSF) (R&D Systems, Inc., Minneapolis, USA), incubated for 10 days in an incubator at 37 °C and 5% CO_2_. On day 3, 10 mL of DC-medium containing rmGM-CSF (20 ng/mL) were added to the culture. In the subsequent media changes on days 6 and 8, half of the culture supernatants were renewed. After 10 days of cultivation, the macrophages (adherent to the ground) were separated from the DCs in suspension and both cells were used for the experiments.

### 4.6. Cell Viability Assay

The CellTiter 96^®^ AQ_ueous_ One Solution Cell Proliferation Assay (MTS; Promega GmbH, Walldorf, Germany) was used to determine cell viability. For the treatment, the MNPSNPs (2 kDa mPEG, 5 kDa mPEG, 10 kDa mPEG-FITC) were separated by a 10-min ultrasonic treatment in sterile *aqua bidestillata*, mixing the suspension for 5 s after 5 min (2500 rpm). Then, dilution to 1500 µg/mL, 500 µg/mL, and 150 µg/mL was followed before adding the MNPSNPs to the maintenance medium at 10% each. A similar procedure was followed for all MNPSNP-treatments described below, unless otherwise stated. In addition, a control with 10% *aqua bidestillata* in the medium and a negative control with 5% DMSO were included. DCs were treated at a cell density of 50,000 cells/well for 24 and 48 h on day 9 after isolation, followed by adding the MTS substance (1:6) for 1 h, and the measurement of the absorbance at 490 nm in the MRX Reader (Dynatech Medical Products Limited, Guernsey, Great Britain). The macrophages were seeded with the same number of cells on day 10 and treated on day 13 of the culture. The primary cultures were performed with three to four technical replicates.

### 4.7. Determination of Cytokine Release

The release of the cytokines TNF-α and IL-6 in the cell culture supernatants was examined using the immunological detection method of the ELISA. The manufacturer’s specifications (R&D Systems, Inc., Minneapolis, MN, USA) were followed during implementation. The 96-well cell culture plates (cf. 4.6.) were mixed with 150 µg/mL, 50 µg/mL, and 15 µg/mL of the MNPSNPs to be examined (2 kDa mPEG, 5 kDa mPEG, and 10 kDa mPEG-FITC) and 1 µg/mL medium of a lipopolysaccharide (LPS O127:B8, positive control) from *E. coli*. A control with 10% *aqua bidestillata* in the maintenance medium was also included. After the incubation for 24 and 48 h, the supernatants were pooled in reaction tubes and stored at −20 °C, protected from light, until analysis. The DCs were removed from the supernatants by centrifugation (300× *g*, 10 min, RT) before freezing.

### 4.8. Measurement of Reactive Oxygen Species

For measuring ROS after contact with various MNPSNPs, the DCs were separated from the adherently growing macrophages on day 10. After a washing step with 5 mL PBS, the DCs were resuspended in phenol red-free RPMI-1640 with 10% FCS and 100,000 cells/well in a 96-well cell culture plate with black walls (µCLEAR^®^, Greiner Bio-One GmbH, Frickenhausen, Germany). The cells were treated in a final concentration of 150 μg/mL of the MNPSNPs (2 kDa, 5 kDa, and 10 kDa mPEG) or by RPMI-1640 with 10% *aqua bidestillata* (control). After 20 h of incubation in the incubator (37 °C, 5% CO_2_), the measurement was performed using the DCFDA/H_2_DCFDA—Cellular ROS Assay Kit from Abcam (Cambridge, United Kingdom). Each well was covered with 100 µL 2’,7’ dichlorofluorescin diacetate (DCFDA, 40 µM) and 20 µL H_2_O_2_ (1 mM, Carl Roth GmbH & Co. KG, Karlsruhe, Germany) was applied as a positive control. Cells were incubated for 4 h in the incubator before green fluorescence was quantified at 485 nm excitation and 520 nm emission by a fluorescence plate reader (FLUOstar^®^ OPTIMA, BMG LABTECH GmbH, Ortenberg, Germany).

To detect ROS in primary murine macrophages, the petri dish was first washed with PBS on day 10 of generation. The cells were then scraped into phenol red-free RPMI-1640 with 10% FCS, centrifuged at 500× *g* for 5 min at RT, and seeded with 50,000 cells/well on a 96-well cell culture plate with black walls. The treatment and incubation took place on day 13 as described above. The ROS in primary, murine DCs, and macrophages were quantified in two technical replicates each.

### 4.9. Migration Assay of Dendritic Cells

DCs were used on day 10 for the transwell migration test in a Boyden chamber (modified from Bäumer et al. (2008)) [59]. Therefore, a 6-well cell culture plate was used and filled with 1 mL DC-medium per well. The DC-medium was either mixed with 150 µg/mL of MNPSNPs (2 kDa mPEG, 5 kDa mPEG, and 10 kDa mPEG-FITC), with 10% *aqua bidestillata* or with 50 ng/mL Macrophage Inflammatory Protein 3β (MIP-3β, Bio-Techne Corporation, Minneapolis, MN, USA). An insert with a pore size of 8 µm (Sarstedt AG & Co. KG, Nümbrecht, Germany) was placed in each well, the bottom of which was flushed with the liquid level in the well. Then, in each insert, 500,000 cells were added (in 300 µL medium). After an incubation time of 90 min in the incubator (37 °C, 5% CO_2_), the cells from the wells were centrifuged (1000× *g*, 10 min, RT) and, after removing the supernatants, resuspended in 100 µL DC-medium. Cell amount was microscopically determined.

### 4.10. Evaluation of the Immune Competence of Primary Macrophages

The macrophages were examined for their ability to inactivate intracellular *E. coli* after contact with the MNPSNPs (modified from Roth et al. (2015) [60]). Macrophages adhering to the petri dish on day 10 of the culture were scraped into 5 mL of antibiotic-free RPMI-1640 medium after a rinsing step with PBS. After centrifugation (500× *g*, 5 min, RT), 200,000 cells/well were placed in a 24-well cell culture plate.

On day 13, the macrophages were treated in four technical replicates with 400 µL of a 150 µg/mL concentrated nanoparticle suspension (2 kDa mPEG, 5 kDa mPEG, and 10 kDa mPEG-FITC) or 10% *aqua bidestillata* in RPMI medium. After incubation for 24 h, the supernatant was removed, and the cells were infected with 500 μL of an *E. coli* suspension. For this purpose, colony material of the reference strain ATCC^®^ 25922™ was adjusted to an optical density of 2.0 at 600 nm in NaCl solution using a photometer (CO8000 Cell Density Meter; Walden Precision Apparatus Ltd., Cambridge, United Kingdom) and then diluted 1:100 in M9 medium. After 2 h of incubation (37 °C, 5% CO*_2_*), the cells were washed once with 500 µL of M9 and then treated with 500 µL of an enrofloxacin solution (0.1 mg/mL M9). The antibiosis was incubated for 1 h in the incubator before the supernatant was replaced with RPMI medium. After 24 h of incubation (37 °C, 5% CO_2_), the supernatants were removed from the wells and the macrophages scraped into 200 µL RPMI medium. The cells were then centrifuged in reaction tubes (12,000× *g*, 5 min, RT) and the cell pellets were resuspended in 500 μL of a 1% sodium dodecyl sulfate (SDS) solution in PBS using a syringe. One hundred microliters of the obtained suspension were streaked onto Columbia agar (Oxoid Deutschland GmbH, Wesel, Germany) and incubated at 37 °C for 24 h. The colony-forming units (CFU) were then determined up to a maximum of 300. The viability of the macrophages was verified at the time of scraping in a parallel setup using MTS.

### 4.11. In Vitro Cellular Uptake Assay

To investigate the internalization of MNPSNPs, macrophages were used on day 10 after isolation. For this purpose, the wells of a 24-well cell culture plate were equipped with cover slips (Ø 12 mm, VWR International GmbH, Darmstadt, Germany) and were then coated with 200 µL poly-L-lysine (150–300 kDa, 0.01% in H_2_O, Sigma-Aldrich Co., St. Louis, USA) at RT. The coverslips were washed three times with 1 mL sterile *aqua bidestillata* and dried for 2 h before macrophages were seeded at a density of 200,000 cells/well. On day 13, FITC functionalized nanoparticles (2 kDa, 5 kDa, and 10 kDa mPEG-FITC) were used after resuspending in phenol red-free RPMI medium. After the respective incubation times (5, 10, 20, and 90 min) at 37 °C and 5% CO_2_, the unabsorbed MNPSNPs were removed from the wells by rinsing twice with 500 µL Hank’s balanced salt solution (HBSS, Gibco Life Technologies Corporation, Grand Island, USA). The macrophages were then fixed with 300 μL of 4% paraformaldehyde (Carl Roth GmbH & Co. KG, Karlsruhe, Germany) in PBS for 15 min in the incubator (37 °C, 5% CO_2_). After washing three times with 300 μL HBSS, the cells were permeabilized with 0.2% Triton™X-100 (300 μL) for 5 min at RT, protected from the light. After two washing steps (300 µL HBSS), the coverslips were placed on parafilm and incubated with 30 µL of the Alexa Fluor^TM^ 568 Phalloidin (1:100 in HBSS, Invitrogen™ Life Technologies Corporation, Waltham, MA, USA) for 1 h at RT, protected from the light. The coverslips were then washed twice (300 µL HBSS) before being immersed in *aqua bidestillata* and then covered with the cells facing down onto 7 µL Antifade Mountant with DAPI (ProLong^®^ Gold, Cell Signaling Technology Inc., Danvers, MA, USA).

For quantification of the phagocytosis, fluorescence images of the XY-plane were taken with a 63 × 1.4 oil immersion objective (Leica TCS SP5, Leica Microsystems GmbH, Wetzlar, Germany). The FITC-labeled MNPSNPs were detected at an excitation (Ex) of 488 nm and emission (Em) of 520 nm, while the nucleus (DAPI) was detected at Ex 405 nm, Em 470 nm and the cytoskeleton (phalloidin) was detected at Ex 570 nm, Em 603 nm. The images were then analyzed using the image processing program ImageJ (ImageJ 1.51q, Wayne Rasband, National Institutes of Health, Bethesda, MD, USA). For this purpose, the cell interior was first drawn using freehand selection within the phalloidin-stained cell surface. Afterwards, the cell outline was transferred to the green channel for subsequent measurement of the enclosed area (Area) and its fluorescence intensity (IntDen). In addition, background fluorescence was determined. Then, the intracellular fluorescence was corrected against the averaged fluorescence (mean) of the background images. Resulting values that were negative due to subtraction of background, were interpreted and presented as zero.

### 4.12. Statistical Analysis

The treatment groups were checked for statistically significant differences against the control using “MIXED” procedure in SAS software SAS^®^ Enterprise Guide 7.15 (SAS Institute Inc., Cary, NC, USA). Groups were considered the main factor of influence for the fixed effect model and “Between-and-within-subject” was used for degrees of freedom. The treatment groups were defined as repeated effect in the subject animal (n = 1). Thereby an unstructured covariance structure was chosen and corrected by Bonferroni post-hoc test.

In experiments with multiple concentrations for one type of MNPSNP, the data were analyzed separately for each PEGylation with the respective positive controls. This resulted in four comparisons per calculation respectively two comparisons for the ROS experiments in DCs. Migration assay and infection trials were checked in one step including all MNPSNPs and positive control. Results of *p* ≤ 0.05 were considered statistically significant: * *p* ≤ 0.05, ** *p* ≤ 0.01; *** *p* ≤ 0.001.

## 5. Conclusions

In this study, we succeeded in functionalizing MNPSNPs with different mPEG chains to investigate them in contact with immune cells. An adequate biocompatibility of all tested MNPSNPs could be confirmed in primary murine DCs and macrophages. The cells did not show increased immune responses to the particles while maintaining their immune competence. Although increased ROS with larger PEG chains were detected, no adverse effects on macrophage viability or functionality were found. There were no differences found for the various mPEG chains.

Furthermore, the studies on the internalization of MNPSNPs failed to show the expected effect of delayed uptake with increasing mPEG mass. MNPSNPs were internalized quickly within minutes and at a comparable amount for all PEGylations. Therefore, further optimization of surface modification would be required for effective use as a drug delivery system.

## Figures and Tables

**Figure 1 ijms-24-02565-f001:**
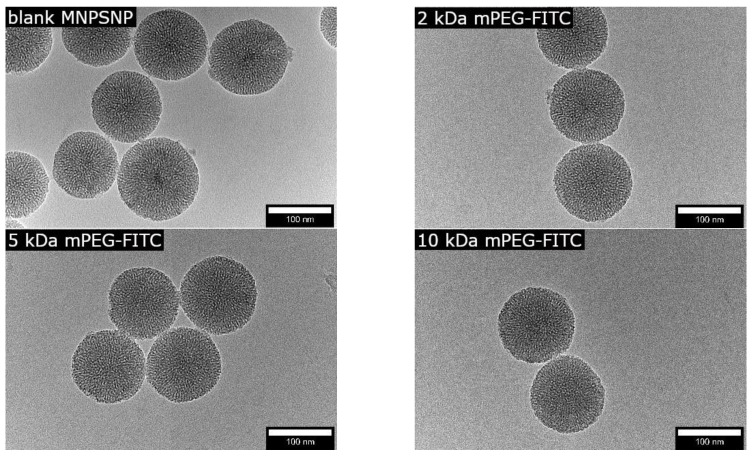
TEM images of unfunctionalized and functionalized MNPSNPs with 2 kDa mPEG-FITC, 5 kDa mPEG-FITC, and 10 kDa mPEG-FITC; FITC—Fluorescein-isothiocyanat Isomer I.

**Figure 2 ijms-24-02565-f002:**
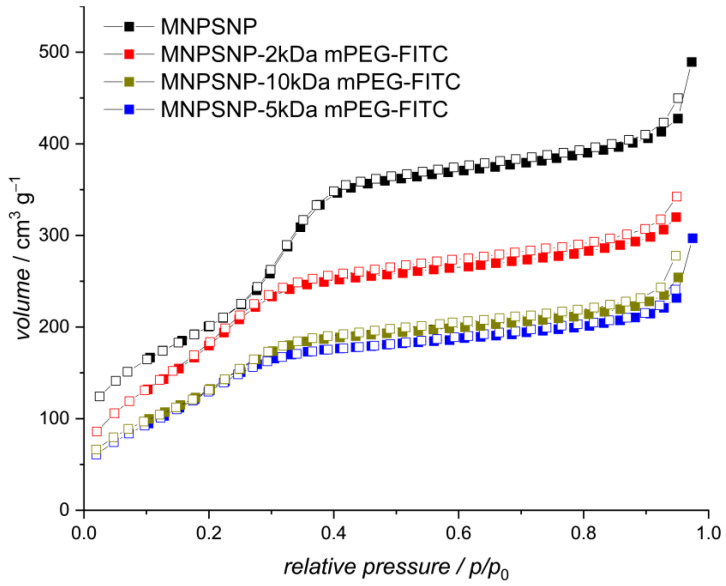
Nitrogen physisorption isotherms of unfunctionalized and functionalized MNPSNPs; empty squares: desorption branch, filled squares: adsorption branch.

**Figure 3 ijms-24-02565-f003:**
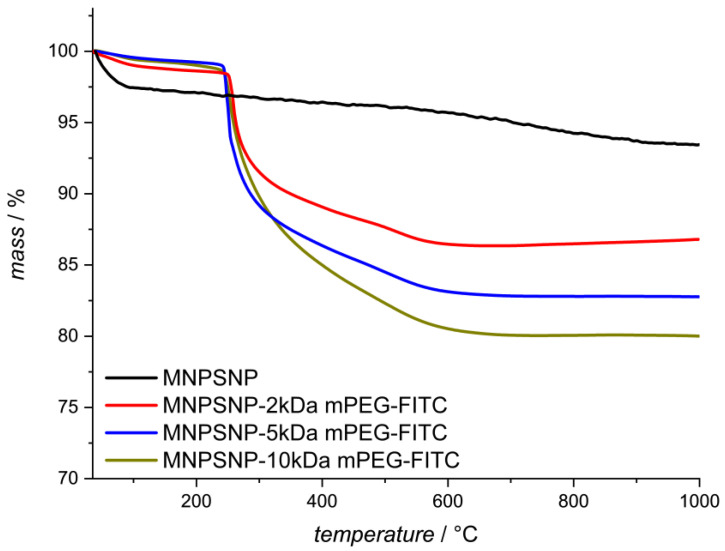
Thermogravimetric curves of unfunctionalized and functionalized MNPSNPs.

**Figure 4 ijms-24-02565-f004:**
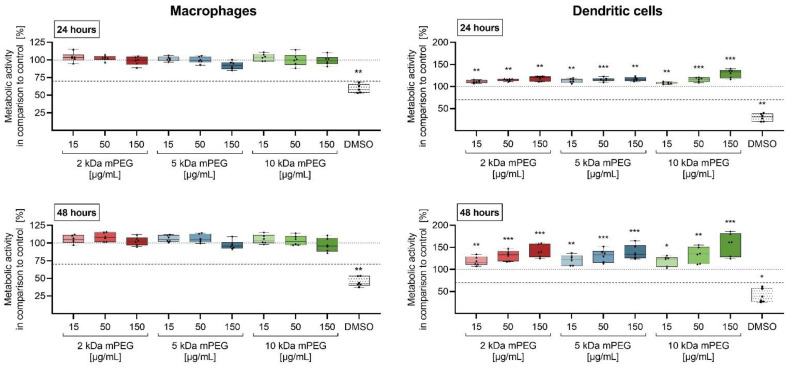
Metabolic activity of macrophages and dendritic cells after incubation with different MNPSNPs for 24 and 48 h in comparison to the control. Data are shown as boxplot with min/max-whisker. The dotted line represents the control (10% *aqua bidestillata*; 100%); the dashed line represents the 70% viability limit according to ISO 10993-5;2009-06 [29]. One-way analysis of variance (ANOVA) with Bonferroni post-hoc test; * *p* ≤ 0.05, ** *p* ≤ 0.01 *** *p* ≤ 0.001; n = 6 each group; asterisks indicate significant differences versus control.

**Figure 5 ijms-24-02565-f005:**
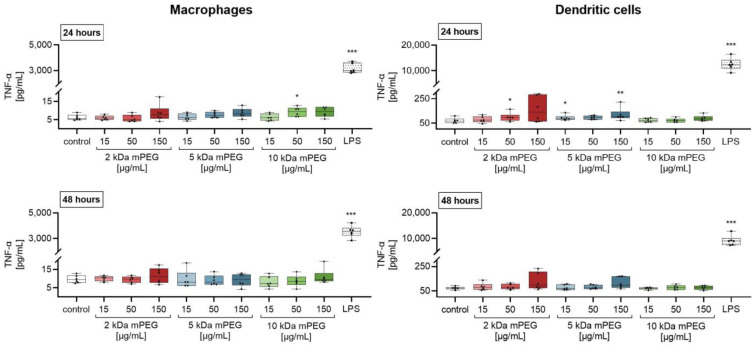
TNF-α concentration (pg/mL) in the supernatants of macrophages and dendritic cells after incubation with different MNPSNPs for 24 and 48 h. Data are shown as boxplot with min/max-whisker. Control equals 10% *aqua bidestillata* in the medium. One-way ANOVA with Bonferroni post-hoc test; * *p* ≤ 0.05, ** *p* ≤ 0.01; *** *p* ≤ 0.001; n = 6 each group; asterisks indicate significant differences versus control.

**Figure 6 ijms-24-02565-f006:**
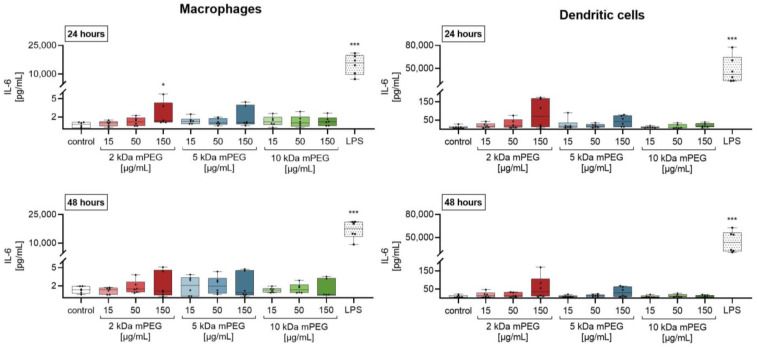
IL-6 concentration (pg/mL) in the supernatants of macrophages and dendritic cells after incubation with different MNPSNPs for 24 and 48 h. Data are shown as boxplot with min/max-whisker. Control equals 10% *aqua bidestillata* in the medium. One-way ANOVA with Bonferroni post-hoc test; * *p* ≤ 0.05; *** *p* ≤ 0.001; n = 6 each group; asterisks indicate significant differences versus control.

**Figure 7 ijms-24-02565-f007:**
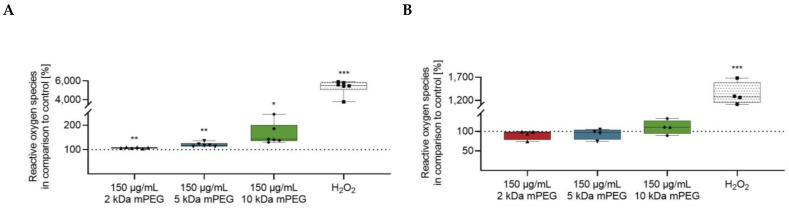
(**A**,**B**) Reactive oxygen species in macrophages (**A**, n = 6 each group) and dendritic cells (**B**, n = 4 each group) after incubation with different MNPSNPs over 24 h in comparison to the control (10% *aqua bidestillata;* dotted line, 100%). (**C**), Migrated dendritic cells (n = 6 each group) in relation to the control (10% *aqua bidestillata*; dotted line, 100%) after incubation with different MNPSNPs. (**D**) Colony forming units of *E. coli* in the cell lysate of MNPSNP-treated macrophages (n = 6 each group). Control equals 10% *aqua bidestillata* in the medium. (**E**) Corrected total intracellular fluorescence of the different MNPSNPs in 6–10 macrophages after incubation for 5-, 10-, 20-, and 90-min. Data are shown as boxplot with min/max-whisker. (**A**–**D**) One-way ANOVA with Bonferroni post-hoc test; * *p* ≤ 0.05, ** *p* ≤ 0.01; *** *p* ≤ 0.001; asterisks indicate significant differences versus control.

**Table 1 ijms-24-02565-t001:** Calculated values for BET surface (BET), pore diameter (*d*), pore volume (*V*), and zeta potential (ZP) of MNPSNPs and functionalized MNPSNPs.

Functionalization	Blank MNPSNP	2 kDa mPEG-FITC	5 kDa mPEG-FITC	10 kDa mPEG-FITC
BET [m^2^·g^−1^]	910	730	610	640
*d* [nm]	3.6	3.4	3.0	3.4
*V* [cm^3^ g^−1^]	0.7	0.5	0.3	0.4
ZP [mV]	−30	−38	−35	−36

## Data Availability

The datasets used and/or analyzed during the current study are available from the corresponding author on reasonable request.

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
