# Peer review of "Influence of PEG Chain Length of Functionalized Magnetic Nanoparticles on the Cytocompatibility and Immune Competence of Primary Murine Macrophages and Dendritic Cells"

_ijms, 2023, doi:10.3390/ijms24032565_

Round 1
Reviewer 1 Report
Dear Editor,
The present article investigates of Influence of PEGylated MNPs on the cytocompatibility and immune competence of some cells. The article has many positive points. The article is well-written and well-detailed. The sections on synthesis and characterization of the produced nanostructures, which were in my field, did not contain any special problems and are suitable for this article. however, below I will mention some of the problems seen in the MS text.
1. minor:
-Please correct grammar and spelling errors throughout the MS. For example, “chain lengths on {the} competence of immune“
-There are many Figures. Transfer some of them to the supplementary data and combine figures 7-8-9 and 10.
2. Major:
-Please recheck the speed of temperature increase in line 423
-Add the solvent percentage composition in line 429
Author Response
Dear Reviewer,
Thank you very much for your valuable comments on our manuscript. We are very happy about your improvments and addressed each point of you (please see below).
-Please correct grammar and spelling errors throughout the MS. For example, “chain lengths on {the} competence of immune“
Answer: We checked the whole manuscript for errors and highlighted them in the tracked mode.
-There are many Figures. Transfer some of them to the supplementary data and combine figures 7-8-9 and 10.
Answer: We combined figures 7-10 (lines 211-212; now there is a figure 7 A-E) and put the pictures 11 and 12 in the supplemented data (lines 689ff).
-Please recheck the speed of temperature increase in line 423
Answer: The temperature increase is correct as written in the manuscript (line 442): „The surfactant was then removed by calcination at 550 °C for 5 hours with a heating rate of 1 °C/minute.“
-Add the solvent percentage composition in line 429
Answer (line 447-449): We changed the sentence to add the information about the composition: „In general, the mPEG derivative was dissolved in a mixture of THF and 0.54 M NaOH (2 kDa mPEG: 10 mL THF, 40 ml 0.54 M NaOH (1:4), 5 kDa mPEG: 1:2, 10 kDa mPEG: 1:0.3).“
Reviewer 2 Report
Authors should comply the following.
Authors have prepared magnetite NPs, then, porous silica nanoparticles are condensed over it. Interestingly, authors did not perform any characterization of formed magnetite NPs, which is required, atleast particle size determination. Particle size of magnetic NPs, magnetic nanoparticles incorporated porous silica NPs and PEGylated NPS should be performed.
Authors should perform Energy dispersive spectroscopy to confirm in presence of magnetite NPs into the porous silica nanoparticles.
Authors stated, ‘Investigations on the uptake of MNPSNPs revealed no differences in the onset of internalization and the ingested amount’. How did the authors calculate the ingested amount. If not rewrite the statement.
Incorporate scale bar in Fig 1.
Manuscript must be checked for the typos.
Author Response
Dear Reviewer,
Thank you very much for your valuable comments on our manuscript. We are very happy about your improvments and addressed each point of you (please see below).
Authors have prepared magnetite NPs, then, porous silica nanoparticles are condensed over it. Interestingly, authors did not perform any characterization of formed magnetite NPs, which is required, atleast particle size determination. Particle size of magnetic NPs, magnetic nanoparticles incorporated porous silica NPs and PEGylated NPS should be performed.
Answer: In 2.1. sizes of magnetite cores used for synthesis of silica coated NPs and (unfunctionalized) MNPSNPs were descripted as determined in TEM: “The synthesized spherical MNPSNPs showed a uniform size distribution with a diameter of 123 ± 16 nm as determined by transmission electron microscopy (TEM). Dark spots could be seen within the MNPSNPs, which represent the magnetite cores. These cores showed a diameter of 10 2 nm. The shape and pore system of the particles after functionalization were preserved, as shown in Figure 1.“ (lines 70-74)
Authors should perform Energy dispersive spectroscopy to confirm in presence of magnetite NPs into the porous silica nanoparticles.
Answer: In section 2.1. dark spots can be seen in the MNPSNPs, which represent the magnetite cores (Figure 1). This is also described in line 71-72 after usage of transmission electron microscopy (TEM). Thus, no Energy dispersive spectroscopy was further used, since no additional information would be obtained.
Authors stated, ‘Investigations on the uptake of MNPSNPs revealed no differences in the onset of internalization and the ingested amount’. How did the authors calculate the ingested amount. If not rewrite the statement.
Answer: This section is rewritten as followed: “onset of internalization and the intensity of intracellular fluorescence“ (lines 31-32)
Incorporate scale bar in Fig 1.
Answer: The scale bar is inserted in Figure 1 (line75)
Manuscript must be checked for the typos.
Answer: We checked the whole manuscript for errors and highlighted them in the tracked mode.